# Prediction of highly stable 2D carbon allotropes based on azulenoid kekulene

Zhenzhe Zhang [1], Hanh D. M. Pham [1], Dmytro F. Perepichka [1] ✉ & Rustam Z. Khaliullin [1] ✉

Despite enormous interest in two-dimensional (2D) carbon allotropes, discovering stable 2D carbon structures with practically useful electronic properties presents a significant challenge. Computational modeling in this work shows that fusing azulene-derived macrocycles – azulenoid kekulenes (AK) – into graphene leads to the most stable 2D carbon allotropes reported to date, excluding graphene. Density functional theory predicts that placing the AK units in appropriate relative positions in the graphene lattice opens the 0.54 eV electronic bandgap and leads to the appearance of the remarkable 0.80 eV secondary gap between conduction bands – a feature that is rare in 2D carbon allotropes but is known to enhance light absorption and emission in 3D semiconductors. Among porous AK structures, one material stands out as a stable narrow-multigap (0.36 and 0.56 eV) semiconductor with light charge carriers ($m_e = 0.17\,m_0$, $m_h = 0.19\,m_0$), whereas its boron nitride analog is a wide-multigap (1.51 and 0.82 eV) semiconductor with light carriers ($m_e = 0.39\,m_0$, $m_h = 0.32\,m_0$). The multigap engineering strategy proposed here can be applied to other carbon nanostructures creating novel 2D materials for electronic and optoelectronic applications.

The extended network of conjugated $\pi$-bonds in 2D carbon nanomaterials, such as graphene, nanotubes, nanosheets, nanoribbons and 2D polymers[1–5] gives rise to their unique properties and make them promising candidates for manufacturing electronic and optoelectronic devices. For example, graphene exhibits a highly-symmetric perfectly hexagonal pattern of $\pi$-conjugated bonds that endows it with the electronic Dirac cones and nearly massless charge carriers[1,2]. However, enormous research efforts have been dedicated to modifying the structure of graphene through doping, lattice strain, chemical substitution, and introduction of pores[6,7] with the aim of converting semimetallic graphene into a semiconductor while retaining the same high mobility of its charge carriers.

One of the most dramatic approaches to alter the electronic properties of the perfectly hexagonal graphene is to rearrange the bonding between $sp^2$-hybridized carbon atoms, creating 2D $\pi$-conjugated carbon allotropes[8]. Computer modeling has shown that introducing azulene structural motifs (i.e., fused pentagonal and heptagonal rings) into the perfect honeycomb lattice of graphene leads to numerous stable graphene allotropes (Supplementary Fig. 1): H567[9], R57b[9], Pza-$C_{10}$[10], pentaheptite[11], phagraphene[12], $\psi$-graphene[13], azugraphene[14], SW-graphene[15], PAI-graphene[16] and PHH-graphene[17]. These structures correspond to local energy minima on the potential energy surface and are expected to be stable at least at low temperatures. Indeed, while few 2D carbon allotropes have been synthesized[18,19], the synthetic accessibility has been demonstrated for azulene-containing 2D fragments[20] and several azulene-containing 2D extended structures, such as phagraphene nanoribbons[21], TPH-graphene nanoribbons[21], and monolayer amorphous carbon[22]. The existence of point Stone-Wales defects in graphene, which can be viewed as two azulene units fused head-to-tail[23], and pentagon-heptagon line defects on graphene grain boundaries[24] are also indicative of the stability of azulene structural motifs.

In addition to producing stable structures, incorporation of azulene motifs into graphene is expected to alter its properties because

[1]Department of Chemistry, McGill University, 801 Sherbrooke St West, Montreal H3A 0B8 QC, Canada. ✉e-mail: dmytro.perepichka@mcgill.ca; rustam.khaliullin@mcgill.ca

azulene molecule with its fused 5- and 7-atom carbon rings exhibits electronic properties that are noticeably different from those of its hexagonal isomer naphthalene. Unlike naphthalene, azulene molecule possesses a large dipole moment of 1.08 D that arises from its electron-rich pentagonal and electron-poor heptagonal rings. This non-uniform charge distribution results in a reduced HOMO-LUMO gap of azulene compared to that of naphthalene. Unlike most molecules that undergo a radiative transition from the lowest excited state to the ground state, excited azulene molecules emit photons mostly from the second excited state[25]. This unusual behavior is a result of the uncommonly large energy gap between the first and the second excited states of azulene[25]. Unusual electronic properties have made azulene and its derivatives the focus of intense research in the field of organic and material chemistry[25].

Several computational studies have shown that azulene-based 2D carbon allotropes can become metals[11,13,16,26,27], while others remain Dirac-cone semimetals[12,14,15,17]. Unfortunately, despite unique electronic features of azulene and promising stability of azulene-based 2D allotropes, none of them possesses electronic properties necessary for the creation of semiconductor devices. The only semiconducting 2D carbon allotrope predicted to have high carrier mobility, Me-graphene[28], is not azulene-derived and predicted to be substantially less stable.

In this work, we report a convenient protocol to design new 2D π-conjugated carbon nanostructures containing various patterns of azulene units. DFT shows that these structures are among the most stable 2D carbon allotropes proposed to date. The electronic structure modeling reveals that several of these materials are semiconductors with light charge carriers and, remarkably, a noticeable secondary energy gap between conduction states. While such well-separated unoccupied electronic states are a common feature in molecular azulene derivatives, they have not been reported in azulene-based 2D carbon allotropes and are exceedingly rare in 2D carbons.

## Results

### Materials design

The primary building block of new materials is AK[29], a fused aromatic macrocycle composed of six azulene units (Fig. 1a). This molecule is an isomer of kekulene – a polycyclic aromatic hydrocarbon of the circulene family[30]. The primary motivation of choosing AK as the building unit of new materials is its unique geometry. First, AK molecule is

expected to be stable because of its low internal strain (Supplementary Note 1). Second, the directions of the outer C-H bonds of AK suggest that it can be incorporated into the graphene matrix without significant distortion as illustrated in Fig. 1a. Finally, the void in the center of AK has the right geometry and size to accommodate a fragment of graphene structure (coronene) rotated 30° relative to the outer graphene lattice (Fig. 1a).

New 2D hexagonal π-conjugated periodic materials are generated by incorporating the AK unit, stripped of its outer hydrogen atoms, into the graphene lattice. Changing the relative position of the AK units creates numerous porous hydrocarbon materials, called porous azulenoid kekulenes (PAK) here. Another set of π-conjugated non-porous materials is generated from the AK units that contains the inner coronene fragment. These materials are composed of only carbon atoms and, therefore, represent 2D carbon allotropes. They are named AK carbons (AKC).

All new PAK and AKC materials are classified according to the relative position of their periodically repeated AK units, specified by two integer numbers $n$ and $m$. These numbers define the lattice vectors of PAK and AKC materials $\vec{A}_i$ in terms of the lattice vectors of the graphene matrix $\vec{a}_i(G)$ (Fig. 1a):

$$\vec{A}_1 = n\vec{a}_1(G) + m\vec{a}_2(G)$$
$$\vec{A}_2 = -m\vec{a}_1(G) + (n-m)\vec{a}_2(G) \tag{1}$$

These equation describe hexagonal lattices with the 120° angle between the lattice vectors. All new materials are labeled using PAK-$[n,m]$ and AKC-$[n,m]$ nomenclature. An alternative nomenclature proposed for 2D carbon allotropes[31] can also be used to name AKC materials (Supplementary Fig. 3).

Fig. 1 shows the structure of the seven AKC materials with different $[n,m]$ values investigated in this work. Seven PAK materials with the same $[n,m]$ values were also studied. These new materials were compared to an additional set of seven control structures generated by removing the AK units completely from the graphene matrix and capping the dangling bonds with hydrogen atoms. These control materials are porous graphene structures and, therefore, denoted as PG-$[n,m]$ (Fig. 2c).

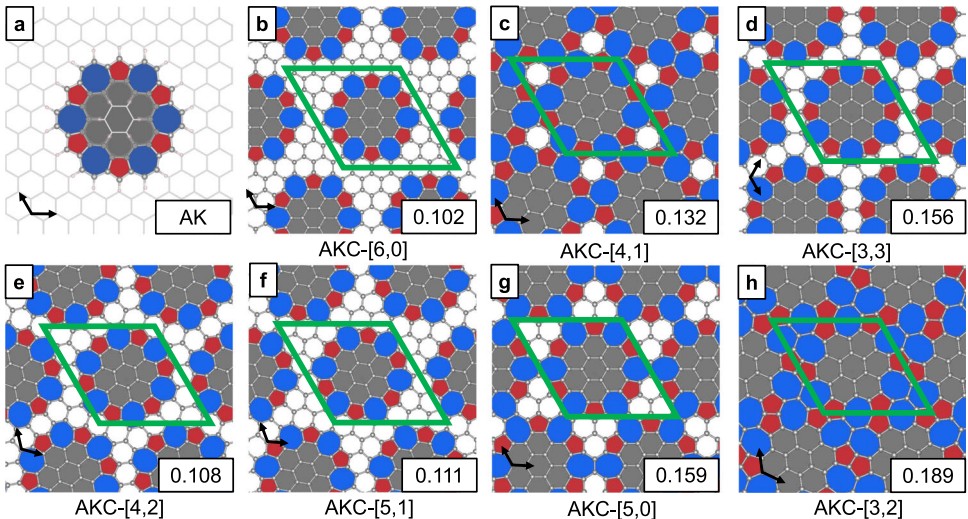

**Fig. 1 | 2D azulenoid kekulene carbons (AKC). a** Azulenoid kekulene (AK) molecule superimposed onto the graphene lattice. **b–d** AKC semiconductors in order of increasing PBE energy. **e–h** AKC semimetals and metals in order of increasing PBE energy. Materials in (**b**)–(**h**) are labeled using AKC-$[n,m]$ nomenclature in Eq. (1). The energy is shown in the bottom right corners in eV/atom above graphene. In all panels, pentagons and heptagons are shown with red and blue colors, respectively. The coronene units inside the AK unit are shown with gray color, whereas the graphene matrix is colored white. The lattice vectors of the graphene matrix are represented by black arrows, whereas the unit cells of the materials are shown in green.

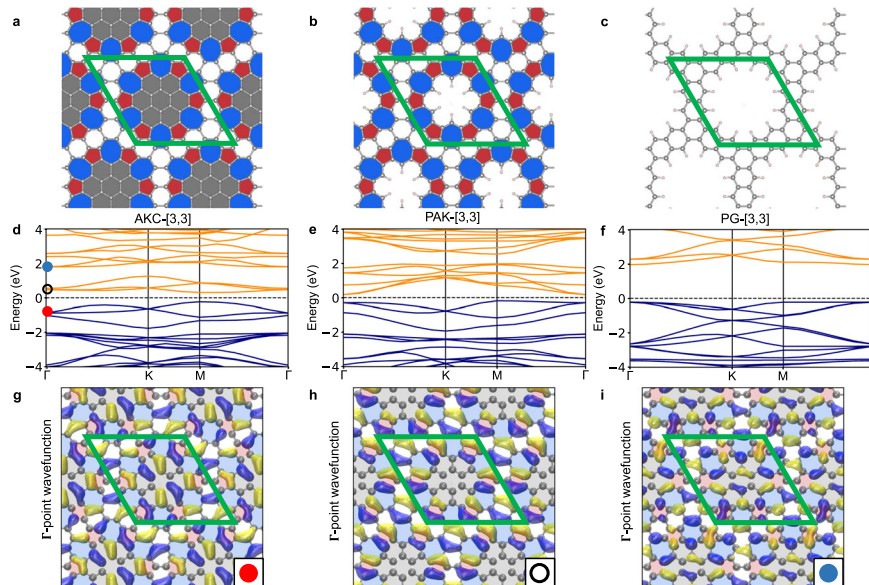

**Fig. 2 | Azulenoid kekulene carbon (AKC) with the corresponding porous azulenoid kekulene (PAK) and porous graphene (PG).** PBE optimized structures of (**a**) AKC-[3,3], (**b**) PAK-[3,3], and (**c**) PG-[3,3] materials. Pentagons and heptagons are shown with red and blue colors, respectively. The coronene units inside the AK unit are shown with gray color, whereas the graphene matrix is colored white. **d**–**f** The corresponding HSE electronic band structure diagrams of the three materials. In the band structure diagrams (**d**–**f**), the blue and orange lines show the valence and conduction bands, respectively. In the band structure diagram of AKC-[3,3] (**d**), the circles mark the location of the orbital diagrams: **g** the valence band, **h** the conduction band immediately above the Fermi level, and **i** the conduction band immediately above the second gap. In (**g**)–(**i**), the yellow and blue colors show positive and negative regions, respectively, of the real electronic wavefunctions at the Γ point. The isosurface value for the wavefunctions is $3 \times 10^{-5}$.

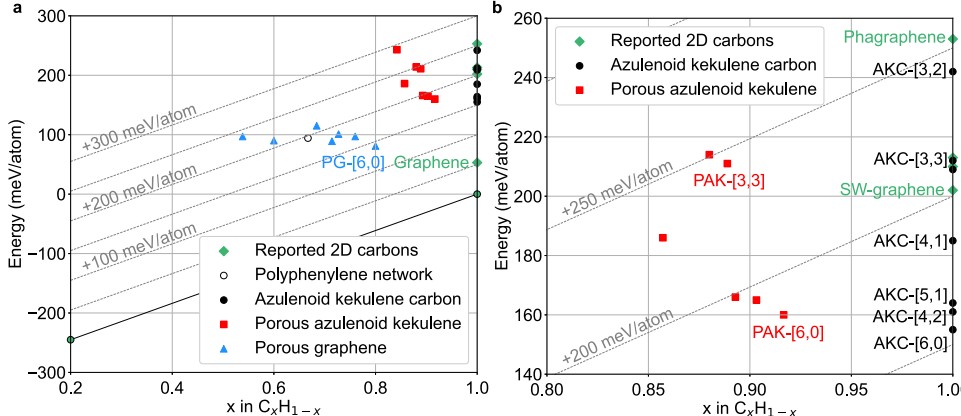

**Fig. 3 | PBE convex hull diagram. a** Azulenoid kekulene carbons (AKC), porous azulenoid kekulenes (PAK), and porous graphenes (PG) on the PBE convex hull diagram for hydrocarbons $C_xH_{1-x}$ defined by the hydrogen molecular crystals, solid methane, and hexagonal graphite. **b** Magnified region of the convex hull diagram showing AKC and PAK materials. The dashed lines show the constant energy levels above the convex hull. Polyphenylene network is a 2D porous graphene that has been synthesized on metal surfaces[33].

## Stability

The structural and electronic properties of all 2D materials were predicted using DFT as described in Methods. The Perdew-Burke-Ernzerhof (PBE) generalized gradient approximation corrected to describe dispersion interactions was used for structure optimization, whereas the Heyd-Scuseria-Ernzerhof (HSE) hybrid functional corrected for dispersion interactions was used to predict electronic properties.

The thermodynamic stability of all newly proposed structures was estimated by calculating their total energy and comparing it to that for the most stable known 2D carbon allotropes and 2D hydrocarbons. In the case of fully optimized unit cells here, the relative energy is an estimate of the zero-temperature zero-pressure enthalpy. Fig. 3 shows the stability of the new 2D materials in the convex hydrocarbon hull

diagram, whereas Table 1 lists the numerical values of the energy above the hull. For the carbon-only AKC materials, the energy above graphene – the most stable 2D allotrope – is also reported in Table 1.

All carbon AKC materials are among the most stable 2D carbon allotropes reported to date. With their energy ranging from 102 MeV/atom above graphene for AKC-[6, 0] to 189 MeV/atom for AKC-[3, 2], three of them are significantly more stable than SW-graphene – the most stable previously reported 2D allotrope with the energy of 149 MeV/atom above graphene (Table 1). Even the least stable AKC material is more stable than one of the most stable previously reported structures – phagraphene (200 MeV/atom above graphene)[12], the nanoribbons of which has been synthesized recently[21]. Furthermore, the computed energy for the experimentally synthesized 2D carbon biphenylene containing 4-, 6-, and 8-atom rings is 445 MeV/atom above

**Table 1 | Electronic structure properties of 2D $\pi$-conjugated materials computed using HSE or PBE functional**

| Material | PBE energy above hull (graphene) (eV/atom) | HSE bandgap (secondary gap) (eV) | Direct or indirect bandgap | HSE effective electron mass ($m_0$) | HSE effective hole mass ($m_0$) |
|---|---|---|---|---|---|
| AKC-[6, 0] | 0.155 (0.102) | 0.83 | I | 1.516 | 0.464 |
| AKC-[4, 2] | 0.161 (0.108) | DC[a] | | | |
| AKC-[5, 1] | 0.164 (0.111) | DC[a] | | | |
| AKC-[4, 1] | 0.185 (0.132) | 0.43 | I | 0.482, 0.756 | 0.225 |
| AKC-[3, 3] | 0.209 (0.156) | 0.54 (0.80) | D (I) | 0.811 | 0.265, 0.476 |
| AKC-[5, 0] | 0.212 (0.159) | M[b] | | | |
| AKC-[3, 2] | 0.242 (0.189) | M[b] | | | |
| SW-graphene[15] | 0.202 (0.149) | PBE: DC[a] | | | |
| Azugraphene[14] | 0.210 (0.157) | PBE: DC[a] | | | |
| $\psi$-graphene[13] | 0.213 (0.16) | PBE/HSE: M[b] | | | |
| Phagraphene[12] | 0.253 (0.20) | PBE: DC[a] | | | |
| PAK-[6, 0] | 0.185 | 0.11 | D | 0.177 | 0.159 |
| PAK-[5, 1] | 0.194 | DC[a] | | | |
| PAK-[4, 2] | 0.199 | DC[a] | | | |
| PAK-[4, 1] | 0.230 | 0.28 | I | 0.433, 0.636 | 0.190 |
| PAK-[3, 3] | 0.245 | 0.36 (0.56) | I (D) | 0.165 | 0.185 |
| PAK-[5, 0] | 0.251 | DC[a] | | | |
| PAK-[3, 2] | 0.291 | DC[a] | | | |
| PG-[6, 0] | 0.142 | 1.73 | D | 0.470, 0.517 | 0.623, 0.691 |
| PG-[5, 1] | 0.170 | DC[a] | | | |
| PG-[3, 3] | 0.176 | 2.18 | D | 0.507 | 0.868 |
| PG-[4, 2] | 0.184 | DC[a] | | | |
| PG-[5, 0] | 0.212 | DC[a] | | | |
| PG-[4, 1] | 0.213 | 3.18 | D | 3.172 | 12.252 |
| PG-[3, 2] | 0.239 | DC[a] | | | |
| BN-AKC-[3, 3] | | 2.07 (0.54) | D (D) | 2.053, 2.368 | 0.465, 0.475 |
| BN-PAK-[3, 3] | | 1.51 (0.82) | D (I) | 0.392, 0.714 | 0.317, 0.593 |
| BN-PG-[3, 3] | | 5.81 | D | 1.244, 1.254 | 0.667, 0.777 |

DC[a] - semimetal with Dirac cones. M[b] - metal with zero bandgap.

Within each family, the materials are listed in order of increasing energyWithin each family, the materials are listed in order of increasing energy.

graphene[32], which is noticeably higher than the energies of the AKC structures proposed here. Interestingly, the AKC materials are found to be among the most stable 2D carbon allotropes even according to an alternative measure of their stability, adjusted to account for the presence of graphene patches in their structure (Supplementary Note 2).

The convex hull diagram in Fig. 3 also indicates that hydrocarbon PAK structures proposed in this work are also expected to be stable. For example, PAK-[6, 0], PAK-[5, 1], and PAK-[4, 2] are predicted to be at least as stable as the polyphenylene network – a 2D porous graphene material that has been synthesized on metal surfaces[33] and is calculated to have the energy of 200 meV/atom above the convex hull, higher than that for the three PAK materials.

In addition to having low energy and thus being relatively stable thermodynamically, experiments and calculations suggest that AK-based materials are expected to be kinetically inert at room temperature if synthesized. For example, the transformation of the graphene-incorporated AK monomer, known as a "flower" defect[34], into pristine graphene is unlikely to occur at room temperature[34,35]. Such a transformation can only be induced by a high-voltage transmission electron microscopy electron beam[35]. Ab initio molecular dynamics (Supplementary Note 3) also shows that AKC-[6,0] maintains its structural integrity when heated from 300 K to 2500 K

over the course of 125 ps. This simulation indicates that there are no low-lying barriers separating the system from more stable graphene-like structures and is in agreement with previous Monte Carlo simulations that show that high temperature is required to convert an AK monomer to pristine graphene[35]. DFT calculations also show that the transformation path from the AK monomer to pristine graphene contains several intermediates with the energy as high as 3 eV above the AK reactant[35]. Additionally, the activation barriers separating these intermediates[35] are expected to be as high as those separating the Stone-Wales defect from pristine graphene[36] because all these transformations occur as a carbon-carbon bond rotation. A DFT study has estimated that the energy barrier for the transformation of a Stone-Wales defect to the perfect graphene lattice is 4.4 eV (244 meV/atom for the 18-atom unit cell)[36]. Such a high barrier explains why the fused pentagonal-heptagonal units in the graphene lattice are kinetically inert and coexist with hexagonal rings[21–23].

## Electronic structure

The presence of pentagonal and heptagonal rings has a profound effect on the electronic properties of the newly proposed structures. The essential descriptors of the electronic properties of the AKC, PAK and PG materials – energy bandgaps and effective mass of charge

carriers – are listed in Table 1 and shown in Fig. 4. The complete electronic band structure diagrams are shown in Supplementary Fig. 6.

The electronic structure of the proposed materials is to a large extent determined by the relative position of AK units in the lattice (Table 1). All [6, 0], [3, 3] and [4, 1] materials including AKC, PAK and PG are semiconductors, while the others – [4, 2], [5, 1], [5, 0] and [3, 2] – are metals or semimetals. Whether the materials are semiconducting or metallic is determined by the electron parity of the nodes of the underlying porous graphene lattice (Supplementary Note 4). Semiconductors are obtained with the nodes of even parity, whereas semimetals or metals are obtained when the nodes are of odd parity. All semimetals exhibit Dirac cones at the κ point of the Brillouin zone because of their fully conjugated hexagonal network π-electrons (Supplementary Fig. 6). While the presence of a Dirac cone implies that the electrical conduction of these materials can be described by the movement of highly mobile massless fermions, all such materials require further gap-opening modifications, chemical or mechanical, to be useful in semiconductor applications. The rest of the discussion will focus on the semiconducting materials.

For the reported 2D semiconductors, the bandgap increases in the PAK, AKC, PG series with the same $[n, m]$ (that is, the same relative positions of the AK units, Fig. 4). For example, the most stable PAK structure PAK-[6, 0] exhibits a bandgap of 0.11 eV, whereas the analogous AKC structure AKC-[6, 0] – also the most stable in its family – possesses a wider bandgap of 0.83 eV. The most stable PG material – PAK-[6, 0] – has an even wider gap of 1.73 eV.

Another important property of a semiconductor, that can be calculated from the curvatures of electronic bands (see Methods) is the effective mass of charge carriers (i.e. electrons and holes). Small effective mass of the carriers indicates that they have a potential to move at high velocities when the material is placed in an electric field. Although the actual mobility of charge carriers can be affected by phonons, temperature, and material impurities, semiconductors with light charge carriers are considered desirable in electronic device engineering as long as they have sufficiently large bandgaps – the range between 0.1 eV and 3.5 eV is appropriate, depending on an application.

The lightest charge carriers are predicted to exist in the PAK-[3, 3] material. The effective masses of electrons and holes are expected to be 0.165 $m_0$ and 0.185 $m_0$, respectively. For comparison, such carriers are expected to be as light as those in silicon (0.16 $m_0$)[37] and phosphorene (0.14 $m_0$)[38] – a single layer of black phosphorous used in transistors, electrodes, and solar cells[39,40]. They are also lighter than highly mobile[41] electrons (0.54 $m_0$) and holes (0.44 $m_0$) in a $MoS_2$ monolayer[42]. However, the carriers in PAK-[3, 3] are expected to be heavier compared to those in 3D germanium (0.04 $m_0$)[43] and 2D

silicene (0.04 $m_0$)[44]. In addition to light carriers, PAK-[3, 3] exhibits the bandgap of 0.36 eV and can be classified as a narrow-band semiconductor, a family of materials with bandgaps below 0.6 eV (e.g. bulk InAs has 0.35 eV bandgap) that find applications as infrared detectors[45], optical[46] and thermal imaging[39] devices.

## Bandgap between conduction states

Several of the proposed materials exhibit an interesting electronic feature – a noticeable energy gap between conduction bands (Fig. 2d, e and Table 1). In the band structure diagram of AKC-[3, 3] with the primary direct bandgap of 0.54 eV, there is another 0.7 eV indirect gap between the top of the third conduction state and the bottom of the fourth conduction state. In the band diagram of the analogous PAK-[3, 3] material with the primary indirect bandgap of 0.36 eV, the 0.5 eV secondary indirect gap is present between the sixth and seventh unoccupied states. In contrast, no secondary gap is present between the conduction states of the corresponding PG-[3, 3] porous graphene structure, indicating that the presence of the AK units is responsible for the appearance of the low-energy conduction bands inside the bandgap (Fig. 2). Visualization of the lowest conduction band of AKC-[3, 3] in Fig. 2g–i indeed confirms that the low-lying conduction states in the AKC-[3, 3] material are primarily localized in the AK units, without noticeable amplitudes in the region of the inner hexagons. In contrast, the conduction band lying above the secondary gap is delocalized over both the AK unit and internal hexagons. The observed effect suggests that incorporating properly spaced AK units with their low-lying unoccupied electronic states (Supplementary Note 5) into the pores of semiconducting structures can introduce disperse π-electron conduction states into semiconductor's gap.

The utility of this band engineering strategy is demonstrated by applying it to a single layer of hexagonal boron nitride[47]. To this end, AK carbon rings were inserted into the pores of the boron nitride analog of PG-[3, 3] – an insulator with the 5.81 eV bandgap (Fig. 5 and Table 1). Fig. 5 shows that this structural modification indeed results in the appearance of intermediate conduction states inside the wide gap of BN-PG-[3, 3]. Since the BN-PG-[3, 3] host material has a wider gap than its carbon version, the resulting boron nitride analogs of AKC-[3, 3] and PAK-[3, 3] also have wider gaps. For example, BN-PAK-[3, 3] exhibits the 1.51 eV direct primary gap and 0.82 eV indirect secondary gap – in the range suitable for multiple semiconductor applications. In addition, the effective mass of electrons and holes for BN-PAK-[3, 3] is noticeably lower than those of the BN-PG-[3, 3] host material (Table 1).

Although the secondary gap resembles the large gap between the two lowest excited states in molecular azulene derivatives, the mere presence of azulene units in a 2D structure is clearly insufficient for the secondary gap to appear. No secondary gaps appear in the band structures of previously reported 2D carbon allotropes that incorporate azulene units including azugraphene[14], phagraphene[12], SW-graphene[15], ψ-graphene[13], C-57 carbon[27] and PAI-graphene[48]. In fact, there is only one 2D carbon allotrope that has been found to exhibit such a secondary gap: twin graphene – a hypothetical double-layer semiconductor that has unrealistically high energy (~800 MeV/atom above graphene) due to its strained out-of-plane bonds[49]. Although high-energy non-planar carbon allotrope penta-graphene[50] and its derivative pentagraphyne[51] also exhibit secondary gaps the stability of these exotic structures has been called into question[52]. Therefore, it is the combination of the symmetry of the AK unit and the position of such units that makes the new 2D carbon-based materials extraordinarily stable and their electronic structure unique.

It is important to note that the secondary gap is encountered in the band diagrams of some 3D materials, which are called multigap, multiband or intermediate band materials[53]. The secondary gap is often engineered by doping a wide-band host material with an element that introduces a delocalized conductive intermediate band between the valence and conduction

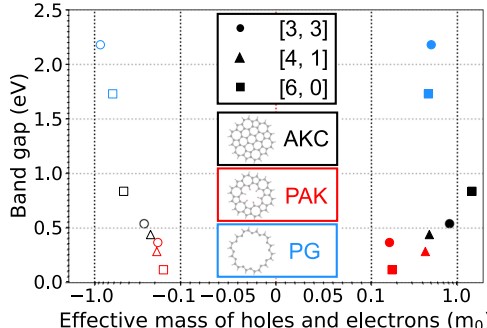

**Fig. 4 | Bandgap and effective carrier mass.** The effective mass of electrons (solid shapes) and holes (with the negative sign, empty shapes) is shown for semiconducting azulenoid kekulene carbons (AKC), porous azulenoid kekulenes (PAK), and porous graphenes (PG). Note that the effective mass x-axis is a linear-logarithmic scale: it is linear in the [−0.1, 0.1] interval and logarithmic outside this interval.

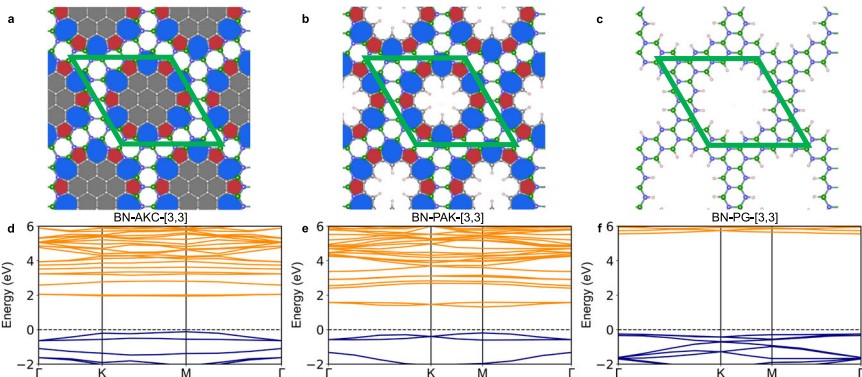

**Fig. 5 | Boron nitride (BN) derivatives of materials based on azulenoid keku-lene.** Structures of (**a**) BN-AKC-[3,3], (**b**) BN-PAC-[3,3], and (**c**) BN-PG-[3,3] materials. Pentagons and heptagons are shown with red and blue colors, respectively. The coronene units inside the AK unit are shown with gray color, whereas the graphene matrix is colored white. **d**−**f** The corresponding HSE band structure diagrams. The boron and nitrogen atoms are shown in green and purple circles, respectively. In the band structure diagrams, the blue and orange lines show the valence and conduction bands, respectively.

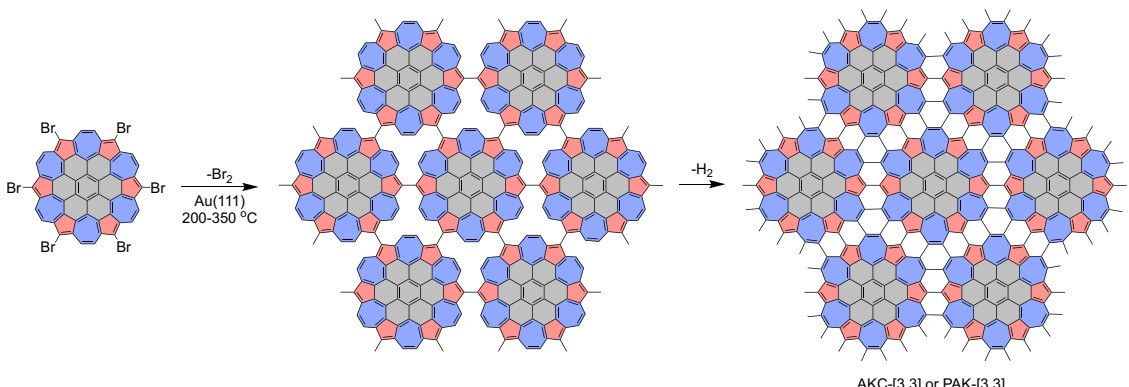

**Fig. 6 | Tentative synthetic route to AKC-[3, 3] and PAK-[3, 3].** The synthesis starts with the Ullmann coupling of the halogenated AK monomers followed by the dehydrogenation reaction. Pentagons and heptagons are shown with red and blue colors, respectively. The six carbon atoms inside each gray region are present in AKC-[3, 3]. In PAK-[3, 3], these carbon atoms are replaced with hydrogen atoms.

bands of the host[54]. Multigap electronic structure has also been predicted for several 2D inorganic semiconductors[55], including monolayer trasnition metal chalcogenides[56].

The presence of multiple gaps instead of one wider gap can be beneficial in emerging semiconductor applications. In photovoltaic applications, the ability of 3D multigap materials to absorb photons of different energies to excite electrons across different bands makes promising candidates for creating full-spectrum solar cells with efficiencies above the Shockley-Queisser limit[53,57]. The presence of multiple gaps can also be beneficial in the light emission processes, such as multicolor emission[58].

### Plausible synthetic routes
To address the question of synthetic plausibility of the proposed structures, it is worth noting that isolated AK units have been previously observed as structural defects in the graphene lattice, obtained by the graphitization of a SiC substrate at 800 °C[34] or by electron beam induced restructuring of graphene[59]. It has also been shown that Er atoms on SiC substrate can act as catalysts resulting in preferential, although still not well controlled, formation of such AK defects[60]. It can be speculated that SiC surfaces with tunable concentration and periodic arrangement of Er (or other lanthanide atoms) could act as templates enabling a spontaneous assembly of ordered AKC lattices of matched periodicity during a standard chemical vapor deposition or during electron-beam induced restructuring.

Alternatively, the future synthesis of AK materials can proceed via on-surface molecular reactions. For example, Ullmann coupling of the hexabrominated AKC or PAK monomers on Au(111) followed by cyclodehydrogenative C−C coupling is expected to lead to AKC-[3,3] and PAK-[3,3] lattices (Fig. 6). A similar approach has been highly successful for the growth of precisely controlled graphene nanoribbons[61,62] as well as 2D π-conjugated polymers[4]. In this approach, the AK unit is "preassembled" by means of organic synthesis of the monomers, and is preserved in polymerization at relatively low temperatures (200 °C–350 °C). The high symmetry of such monomers should minimize the occurrence of lattice defects. We note that no AK-containing monomers have been synthesized to date, although the recent progress in synthesis of related fused oligoazulenes[20] suggests such monomers can be developed using the already established synthetic methodology. Further molecular engineering of the AK monomers can also enable the access to other AKC and PAK homologues. The actual experimental realization of any of such syntheses is a formidable challenge, but we hope that our prediction of the stability and unusual electronic properties of AKC and PAK materials makes this challenge a worthwhile goal.

### Discussion
This work demonstrates that fusing AK units into the graphene lattice produces the most stable 2D carbon allotropes known to date, excluding graphene. DFT calculations predict that, depending on the

positioning of AK units in the lattice, the resulting AK-based 2D materials exhibit properties of semiconductors, Dirac-cone semi-metals, or metals. A proper placement of AK units in graphene lattice can generate semiconducting materials with secondary gaps between their conduction bands. This feature is rare in 2D carbon materials and has been previously predicted only in two high-energy 2D carbons. The secondary gap is known to enhance light absorption and emission properties in 3D materials, making its discovery in stable 2D carbon semiconductors significant for optoelectronic applications of 2D materials.

Among the newly designed materials, PAK-[3, 3] stands out as a stable narrow-multigap semiconductor with light charge carriers. Both bandgaps in this material can be increased by heteroatom substitution to the range desirable for light harvesting applications.

The properties of the materials discussed in this study can be further adjusted using established techniques such as 1D confinement, doping, passivation, and strain. Furthermore, the general approach to engineer stable multi-gap semiconductors has significant implications as it can be applied to other $\pi$-conjugated nanostructures, such as nanotubes, nanosheets, layered and twisted structures, thus expanding the range of materials for electronic and optoelectronic applications.

## Methods

Computational modeling was performed using DFT as implemented in Vienna Ab initio Simulation Package (VASP)[63–66]. The projector augmented wave formalism was used to describe interactions of atomic cores with valence electrons. Electronic states were modeled using a plane wave basis set with the energy cutoff set at 520 eV. The PBE generalized gradient approximation[67] corrected to describe dispersion interactions[68,69] was used for the optimization of lattice vectors and atomic positions.

For the geometry relaxation, the integration over the Brillouin zone was performed using the $11 \times 11 \times 1$ Monkhorst-Pack $k$-point mesh. Convergence criterion for the self-consistent field procedure was set to $10^{-6}$ eV. Atomic positions were optimized until the maximum force on atoms decreased below 0.02 eV Å$^{-1}$. It is important to note that setting pressure to zero ensures that the enthalpy of a 2D material $H = E + pV$ is equal to its energy. The zero pressure approximation is appropriate to describe materials at the pressure of 1 atmosphere. The unit cell of 2D materials was optimized in the two lateral directions with the pressure set to zero. The cell size perpendicular to the plane was fixed at 10 Å, which is sufficiently large to make the effect of interlayer interactions on the computed properties negligibly small (Supplementary Note 6).

Band structure calculations were carried out for the optimized structures with the PBE and HSE exchange-correlation functionals[70,71], both corrected for dispersion interactions. $64 \times 64 \times 1$ and $3 \times 3 \times 1$ Monkhorst-Pack $k$-point meshes were employed in the PBE and HSE calculations, respectively. The band structures of all 2D materials were simulated and plotted along the lines connecting high symmetry points in the Brillouin zone: Γ (0,0, 0.0, 0.0) → K (2/3, 1/3, 0.0) → M (1/2, 0.0, 0.0) → Γ (0,0, 0.0, 0.0).

The effective mass of electrons and holes were calculated from the DFT band structure as curvatures of electronic bands using the `sumo` program[72]. Effective mass for electrons (holes) is reported only for the lowest (highest) energy minimum (maximum) on conduction (valence) bands.

The PBE convex hull for hydrocarbons is defined by the hydrogen molecular crystals in the cubic $\alpha$ N$_2$ structure, solid methane in the silicon tetrafluoride polymorph, and hexagonal (ABAB) graphite.

## Data availability

Atomic coordinates for optimized AKC, PAK and PG structures as well as the initial and final structures from ab initio molecular dynamics simulations for AKC-[6,0] and phagraphene have been deposited in the Figshare database under accession code https://doi.org/10.6084/m9.figshare.25058879[73]. Other data that support the findings of this study are available from the corresponding authors upon request.

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

## Acknowledgements

The research was funded by the Natural Sciences and Engineering Research Council of Canada (NSERC) through Discovery Grant (RGPIN-2016-0505, R.Z.K.) and McGill Sustainability Systems Initiative through Ideas Fund (R.Z.K.). The authors are grateful for computer resources allocated under the CFI John R. Evans Leaders Fund program (R.Z.K.).

## Author contributions

Z.Z. performed calculations. H.D.M.P. performed calculations for molecular systems. Z.Z. and R.Z.K. wrote the manuscript. D.P. edited the manuscript. R.Z.K. and D.P. supervised the project.

## Competing interests

The authors declare no competing interests.
