## [Peer Review File · Nature Communications]

Prediction of Highly Stable 2D Carbon Allotropes Based on Azulenoid KekuleneReviewer #1 (Remarks to the Author):

This paper reports a true cornucopia of new carbon-based structures with unique properties. They are more stable than all other such structures reported so far except for graphene. The authors focus on materials that feature energy bands with large enough band gaps to serve as semiconductors for potential electronic devices. Many of the new materials have a secondary gap in the conduction bands which again enshrine unique properties.

The paper is very well written, with only minor grammatical errors (on several occasions the article "the" should have been "a"). The paper is very much worthy of publications in Nature Communications, but I recommend the authors examine the following comments to improve the manuscript.

1. The most serious comment is a concern about the size of the vacuum region between periodically repeated monolayers. It is smaller than 10 Angstroms (10 Angstroms is actually the cell size in the perpendicular direction). Experience tells me that you need at least 15 Angstroms for good convergence of most properties. The authors simply assert that 10 Angstroms is sufficiently large to make interactions negligibly small. I suggest that they take a few materials and increase the vacuum region to 15 Angstroms and check the convergence of several properties (total energy, band gaps, effective mass).

2. I was at first puzzled by Fig. 2c where the authors plot the positive and negative regions of electronic wave functions because wave functions are generally complex quantities. Then I noticed that they only plot Gamma-point wave functions, which are real. I suggest the authors point that out because it is more common to plot electron densities.

3. The authors say at the bottom of p. 2 that the new materials are semiconductors with high carrier mobility, which made me curious as to what criterion they use for high carrier mobility. On p. 8, they equate low effective masses with high carrier mobility, which is not strictly correct. Mobility is also limited by phonons even in pure materials. MoS₂ is notorious about the effect of phonons on carrier mobility. More importantly mobilities can be reduced dramatically in device channels because of the effect of interfaces (silicon's mobility is reduced by a factor 2 in Si MOSFETs, while in the early days of SiC MOSFETs mobility was clobbered completely by the SiC-SiO₂ interface. I recommend they should talk about small effective masses, which in pure materials in the absence of interfaces is often a major determinant of carrier mobilities.

4. At the top of p. 2, the authors talk about band gaps that are wider than that of graphene. Graphene of course has no band gap.

5. I suggest that the paper would benefit from a section in which the authors discuss the likelihood that the predicted materials can be fabricated. Readers would appreciate the authors' reasoned opinions.

Reviewer #2 (Remarks to the Author):

Z. Zhang and coworkers report theoretical considerations for the design of planar pi-conjugated sp² carbon nanostructures containing various patterns of azulene units (AKC materials). On the basis of DFT calculated ground state energies and transition barriers, the authors claim that these structures are among the most stable two-dimensional carbon allotropes proposed to date. The electronic structure modeling suggests that several of these materials may be semiconductors with high carrier mobility and a noticeable secondary energy gap between conduction states. This is highly interesting and timely work that should be published in Nature Communications, but there are a few points in which the manuscript can be improved, as described below.

While there are various theoretical publications about planar sp² carbon allotropes, there is very little related experimental work, due to the difficult synthesis. In this situation, it can be expected that a theory paper that proposes a new class of planar sp² carbon allotropes fulfills at the least the following two criteria: First, it makes a careful comparison with the few available experimental data and second, it provides suggestions how the proposed new materials could be synthesized. The manuscript under review can be improved in both respects.

For example, the question should be addressed why the azulene-based extended structures should be kinetically stable (i.e., have large activation energies for their transformation into graphene or other sp² carbon networks), when azulene itself is known for its low-barrier spontaneous thermal isomerization to naphthalene. Even more importantly, the only work that actually managed to create extended azulene-based structures (phagraphene, TPH-graphene) with some degree of periodicity, shows that the transformation of the azulene motif into hexagonal rings is indeed a major problem during the synthesis (ref. 18 in the manuscript). Perhaps discrepancies between theory and experiment are due to a structurally ideal system on one side and a defect-rich real system on the other. In any case, it should be acknowledged that the situation in a real-world experiment may be rather different. Interestingly, the authors mention biphenylene network (BPN), another experimentally realized sp² carbon network, for comparison, and emphasize that it has a higher energy per atom than the calculated AKC materials. However, the experimental findings are that BPN is highly stable on a gold surface at elevated temperatures, unlike the phagraphene ribbons. This is also in line with the high stability of BPN found in ab initio molecular dynamics (AIMD) simulations, which show that BPN remains intact up to 4500 K. It is suggested that the authors include barrier calculations such as those in figure 3 for extended BPN and phagraphene in comparison to their proposed materials.

Secondly, the manuscript would greatly benefit from suggestions on synthesizing the proposed structures. Compared to theoretical calculations, practical synthesis attempts of sp² carbon allotropes are currently underrepresented, and additional insights in this area would be highly valuable.

Minor points:

"phagraphene (200meV/atom above graphene), the nanoribbons of which has been synthesized recently [11]." – Ref. [11] does not report the synthesis.

"In addition to being thermodynamically stable, AKC materials..."- The AKC materials are metastable, the only stable allotrope is graphene/graphite.

Reviewer #1 (Remarks to the Author):

This paper reports a true cornucopia of new carbon-based structures with unique properties. They are more stable than all other such structures reported so far except for graphene. The authors focus on materials that feature energy bands with large enough band gaps to serve as semiconductors for potential electronic devices. Many of the new materials have a secondary gap in the conduction bands which again enshrine unique properties.

The paper is very well written, with only minor grammatical errors (on several occasions the article "the" should have been "a"). The paper is very much worthy of publications in Nature Communications, but I recommend the authors examine the following comments to improve the manuscript.

We appreciate Reviewer's encouraging comments and constructive criticism. We have addressed all expressed concerns point-by-point below.

1. The most serious comment is a concern about the size of the vacuum region between periodically repeated monolayers. It is smaller than 10 Angstroms (10 Angstroms is actually the cell size in the perpendicular direction). Experience tells me that you need at least 15 Angstroms for good convergence of most properties. The authors simply assert that 10 Angstroms is sufficiently large to make interactions negligibly small. I suggest that the take a few materials and increase the vacuum region to 15 Angstroms and check the convergence of several properties (total energy, band gaps, effective mass).

To address the concern, we have recalculated the energy and band structures for all AKC, PAK, and PG materials with the interlayer distance of 20 Angstroms and compared results to the previous (10 Angstrom) calculations in Section S7 (Supplementary Information). The new results match the old data. Supplementary Table S2 shows the discrepancy in the energy and band gap is negligible, with the maximum difference of 0.4 meV/atom and 0.003 eV, respectively. These small differences confirm the validity of the model with the 10 Angstrom interlayer distance. The reason for the validity of the 10 Angstrom model is the absence of long-range electrostatic interactions between the layers and small size of carbon atoms. With the electron density being negligibly small at 5 Angstrom from the carbon nuclei, all other interactions between layers separated by 10 Angstrom are nearly zero.

2. I was at first puzzled by Fig. 2c where the authors plot the positive and negative regions of electronic wave functions because wave functions are generally complex quantities. Then I noticed that they only plot Gamma-point wave functions, which are real. I suggest the authors point that out because it is more common to plot electron densities.

We have updated the legend of Fig.2 to clarify that the Γ -point wavefunction is plotted.

3. The authors say at the bottom of p. 2 that the new materials are semiconductors with high carrier mobility, which made me curious as to what criterion they use for high carrier mobility. On p. 8, they equate low effective masses with high carrier mobility, which is not strictly correct. Mobility is also limited by phonons even in pure materials. MoS2 is notorious about the effect of phonons on carrier mobility. More importantly mobilities can be reduced dramatically in device channels because of the effect of interfaces (silicon's mobility is reduced by a factor 2 in Si MOSFETs, while in the early days of SiC MOSFETS mobility was clobbered completely by the

SiC-SiO₂ interface. I recommend they should talk about small effective masses, which in pure materials in the absence of interfaces is often a major determinant of carrier mobilities.

We have followed the reviewer's recommendation and changed the wording throughout the manuscript (highlights, particularly those in page 9 of the corrected version of the manuscript) to clarify that this work estimates only the effective mass of charge carriers, not the carrier mobility. Although the mobility is inversely proportional to the effective mass of the carriers (in a simplified theory), the mobility can indeed be affected by numerous other factors such as phonons, temperature, material impurities, and interfaces.

4. At the top of p. 2, the authors talk about band gaps that are wider than that of graphene. Graphene of course has no band gap.

We have revised the sentence mentioned by the reviewer: "with the aim of converting semimetallic graphene into a semiconductor while retaining the same high mobility of its charge carriers."

In the original version, we meant the near zero band gap of graphene induced by the spin-orbit coupling [<https://doi.org/10.1103/PhysRevLett.95.226801>].

5. I suggest that the paper would benefit from a section in which the authors discuss the likelihood that the predicted materials can be fabricated. Readers would appreciate the authors' reasoned opinions.

An extensive discussion of several plausible synthetic routes to fabricate the new materials has been added as a new section entitled "Plausible synthetic routes" in the revised manuscript (page 11).

Reviewer #2 (Remarks to the Author):

Z. Zhang and coworkers report theoretical considerations for the design of planar pi-conjugated sp² carbon nanostructures containing various patterns of azulene units (AKC materials). On the basis of DFT calculated ground state energies and transition barriers, the authors claim that these structures are among the most stable two-dimensional carbon allotropes proposed to date. The electronic structure modeling suggests that several of these materials may be semiconductors with high carrier mobility and a noticeable secondary energy gap between conduction states. This is highly interesting and timely work that should be published in Nature Communications, but there are a few points in which the manuscript can be improved, as described below.

While there are various theoretical publications about planar sp² carbon allotropes, there is very little related experimental work, due to the difficult synthesis. In this situation, it can be expected that a theory paper that proposes a new class of planar sp² carbon allotropes fulfills at the least the following two criteria: First, it makes a careful comparison with the few available experimental data and second, it provides suggestions how the proposed new materials could be synthesized. The manuscript under review can be improved in both respects.

We are grateful to the reviewer for the positive evaluation of our manuscript and constructive feedback. We have revised our manuscript to address both points as detailed below.

For example, the question should be addressed why the azulene-based extended structures should be kinetically stable (i.e., have large activation energies for their transformation into

graphene or other sp² carbon networks), when azulene itself is known for its low-barrier spontaneous thermal isomerization to naphthalene. Even more importantly, the only work that actually managed to create extended azulene-based structures (phagraphene, TPH-graphene) with some degree of periodicity, shows that the transformation of the azulene motif into hexagonal rings is indeed a major problem during the synthesis (ref. 18 in the manuscript). Perhaps discrepancies between theory and experiment are due to a structurally ideal system on one side and a defect-rich real system on the other. In any case, it should be acknowledged that the situation in a real-world experiment may be rather different. Interestingly, the authors mention biphenylene network (BPN), another experimentally realized sp² carbon network, for comparison, and emphasize that it has a higher energy per atom than the calculated AKC materials. However, the experimental findings are that BPN is highly stable on a gold surface at elevated temperatures, unlike the phagraphene ribbons. This is also in line with the high stability of BPN found in ab initio molecular dynamics (AIMD) simulations, which show that BPN remains intact up to 4500 K. It is suggested that the authors include barrier calculations such as those in figure 3 for extended BPN and phagraphene in comparison to their proposed materials.

We have added an extensive discussion of the kinetic stability of the new structures (see the highlighted paragraph in page 7 of the corrected version of the manuscript) by referring to available experimental and computational evidence of the stability of AK monomers incorporated into the graphene lattice.

Additionally, we have performed AIMD simulations of AKC-[6,0] – one of the new materials – to show that there are no low-lying barriers separating the AKC-[6,0] from more stable graphene-like structures. Following reviewer's recommendation, we also briefly discussed the limitation of AIMD simulations that typically describe defect-free systems.

Finally, we would like to note that Figure 3b includes phagraphene. The new materials are also compared to BPN in the text. It is, however, difficult to include BPN in Figure 3 because it lies significantly higher in energy than phagraphene and the new materials (BPN is 445 meV/atom above graphene).

Secondly, the manuscript would greatly benefit from suggestions on synthesizing the proposed structures. Compared to theoretical calculations, practical synthesis attempts of sp² carbon allotropes are currently underrepresented, and additional insights in this area would be highly valuable.

An extensive discussion of several plausible synthetic procedures to create the new materials has been added as a new section entitled "Plausible synthetic routes" in the revised manuscript.

Minor points:

"phagraphene (200meV/atom above graphene), the nanoribbons of which has been synthesized recently [11]." – Ref. [11] does not report the synthesis.

We apologize for citing the wrong article. We have now included the correct citation to the synthesis of phagraphene nanoribbons.

"In addition to being thermodynamically stable, AKC materials..."- The AKC materials are metastable, the only stable allotrope is graphene/graphite.

We have rephrased this sentence: “In addition to having low energy and thus being relatively stable thermodynamically...”.

We agree with the reviewer’s comment that graphite is the only truly stable thermodynamic state at standard pressure and room temperature, while all other carbon allotropes are metastable. Nevertheless, it is not uncommon to omit the *meta* prefix when the thermodynamic stability of different allotropes is discussed. We hope, however, that our revised sentence avoids the ambiguity.

Reviewer #1 (Remarks to the Author):

The authors made suitable revisions in response to the reviewers' comments. The paper is now suitable for publication, but one needed correction that I just became aware of. In Table 1 and in Fig. 4, hole effective masses are listed as negative. That's not correct. Holes have positive effective masses and carry a positive charge. Check any semiconductor physics book.

Reviewer #2 (Remarks to the Author):

The authors have addressed my comments in a satisfactory way. Publication of this manuscript is recommended.